# Hypoxia and Microvascular Alterations Are Early Predictors of IDH-Mutated Anaplastic Glioma Recurrence

**DOI:** 10.3390/cancers13081797

**Published:** 2021-04-09

**Authors:** Andreas Stadlbauer, Stefan Oberndorfer, Gertraud Heinz, Max Zimmermann, Thomas M. Kinfe, Arnd Doerfler, Michael Buchfelder, Natalia Kremenevski, Franz Marhold

**Affiliations:** 1Institute of Medical Radiology, University Clinic St. Pölten, Karl Landsteiner University of Health Sciences, 3100 St. Pölten, Austria; Gertraud.Heinz@stpoelten.lknoe.at; 2Department of Neurosurgery, Friedrich-Alexander University (FAU) Erlangen-Nürnberg, 91054 Erlangen, Germany; Max.Zimmermann@med.uni-tuebingen.de (M.Z.); thomasmehari.kinfe@uk-erlangen.de (T.M.K.); Michael.Buchfelder@uk-erlangen.de (M.B.); Natalia.Kremenevskaja@uk-erlangen.de (N.K.); 3Department of Neurology, University Clinic of St. Pölten, Karl Landsteiner University of Health Sciences, 3100 St. Pölten, Austria; Stefan.Oberndorfer@stpoelten.lknoe.at; 4Department of Preclinical Imaging and Radiopharmacy, University of Tübingen, 72076 Tübingen, Germany; 5Division of Functional Neurosurgery and Stereotaxy, Friedrich-Alexander University (FAU) Erlangen-Nürnberg, 91054 Erlangen, Germany; 6Department of Neuroradiology, Friedrich-Alexander University (FAU) Erlangen-Nürnberg, 91054 Erlangen, Germany; arnd.doerfler@uk-erlangen.de; 7Department of Neurosurgery, University Clinic of St. Pölten, Karl Landsteiner University of Health Sciences, 3100 St. Pölten, Austria; franz.marhold@stpoelten.lknoe.at

**Keywords:** anaplastic glioma, isocitrate-dehydrogenase, IDH gene mutation, recurrence, hypoxia, neovascularization, treatment failure, vascular cooption, physiological MRI

## Abstract

**Simple Summary:**

Anaplastic gliomas (AGs) are considered the most common and aggressive primary brain tumors of young adults with inevitable recurrence and treatment failure. The aim of this study was to investigate whether the imaging biomarkers of hypoxia, microvascular architecture and neovascularization activity can be of assistance to detect pathophysiological changes in the early developmental stages of isocitrate-dehydrogenase (IDH) mutated AG recurrence. We evaluated 142 physiological magnetic resonance imaging follow-up examinations as a part of the conventional magnetic resonance imaging (MRI) protocol in 60 AG patients after standard therapy. Physiological MRI biomarkers showed intensifying local tissue hypoxia 250 days prior to radiological recurrence with following upregulation of neovascularization activity 50 to 70 days later. Integration of physiological MRI in the monitoring of AG patients may be of clinical significance to make personalized decision of early tumor recurrence without an additional delay for multimodal therapy.

**Abstract:**

Anaplastic gliomas (AG) represents aggressive brain tumors that often affect young adults. Although isocitrate-dehydrogenase (IDH) gene mutation has been identified as a more favorable prognostic factor, most IDH-mutated AG patients are confronted with tumor recurrence. Hence, increased knowledge about pathophysiological precursors of AG recurrence is urgently needed in order to develop precise diagnostic monitoring and tailored therapeutic approaches. In this study, 142 physiological magnetic resonance imaging (phyMRI) follow-up examinations in 60 AG patients after standard therapy were evaluated and magnetic resonance imaging (MRI) biomarker maps for microvascular architecture and perfusion, neovascularization activity, oxygen metabolism, and hypoxia calculated. From these 60 patients, 34 patients developed recurrence of the AG, and 26 patients showed no signs for AG recurrence during the study period. The time courses of MRI biomarker changes were analyzed regarding early pathophysiological alterations over a one-year period before radiological AG recurrence or a one-year period of stable disease for patients without recurrence, respectively. We detected intensifying local tissue hypoxia 250 days prior to radiological recurrence which initiated upregulation of neovascularization activity 50 to 70 days later. These changes were associated with a switch from an avascular infiltrative to a vascularized proliferative phenotype of the tumor cells another 30 days later. The dynamic changes of blood perfusion, microvessel density, neovascularization activity, and oxygen metabolism showed a close physiological interplay in the one-year period prior to radiological recurrence of IDH-mutated AG. These findings may path the wave for implementing both new MR-based imaging modalities for routine follow-up monitoring of AG patients after standard therapy and furthermore may support the development of novel, tailored therapy options in recurrent AG.

## 1. Introduction

Cancers of the central nervous system (CNS) refer to a heterogeneous group of rare tumors [1]. While primary CNS tumors constitute only approximately 3% of the cancer cases worldwide [2], they cause 7% of the years of life lost from cancer before age 70 [3]. Gliomas account for 75% of malignant primary CNS tumors in adults and more than the half of these are glioblastoma [4], associated with a poor outcome in terms of median overall survival of 14–17 months [5] and 5-year survival rate of less than 5% [3]. Anaplastic glioma (AG) as a subtype of aggressive gliomas often affects young adults in the prime of life causing significant disability as well as death [4,6]. Moreover, 70–90% of AG, but only 5–10% of glioblastoma, show a mutation of the isocitrate dehydrogenase (IDH) gene [7], which has been identified as a significant molecular prognostic factor for a relatively more favorable clinical course [8,9]. Despite best-practice multimodal treatment including surgical resection, radiotherapy and chemotherapy [10,11], nearly all IDH-mutated AG patients suffer from incurability in due course. This is mainly due to the diffusively infiltration of glioma cells into the brain parenchyma, which makes a complete surgical extraction of the tumor impossible and inevitably leads to recurrence of the tumor eventually combined with likelihood for malignant transformation to glioblastoma.

There exists sparse evidence addressing the diagnostic management of AG patients after tumor recurrence as most trials focused on heterogeneous patient cohorts including newly diagnosed untreated glioblastomas and AGs, summarized as “malignant” or “high-grade” glioma, with a strong majority of the patients with glioblastoma [12]. Consequently, most conclusions as well as treatment guidelines for AG recurrence are derived from extrapolation of these mixed brain tumor entities with untreated tumors and may not entirely hold true for IDH-mutated AGs, which differ prognostically from glioblastomas. Moreover, pathophysiology of glioma as well as recurrence of AGs has not been well studied in humans so far but mainly in animal models, in particular for recurrence of AGs with IDH-mutation.

It is well documented from these preclinical studies, that the vast majority of glioblastoma cells infiltrate the surrounding brain tissue through the perivascular space along pre-existing blood vessels [13,14]. In the course of this process, which is known as vascular cooption [15], glioblastoma cells disrupt the astrocyte-vasculature interaction [16] and organize themselves into cuffs around microvessels accompanied by upregulation of angiopoietin-2 (ANG-2) in the coopted vessels [15]. This ANG-2 upregulation, however, leads to regression of vascular structures, which is accompanied by a decrease in tissue oxygen tension (intensifying hypoxia) and hypoxia-induced expression of vascular endothelial growth factor (VEGF) [17], ultimately leading to the development of new vessels, a process known as neovascularization [17]. In other words, the tumor switches from an avascular infiltrative to a vascularized proliferative phenotype and either becomes part of the expanding tumor or develops into a new tumor mass [18]. These biological processes that drive progression of untreated glioblastoma are presumably also responsible for the development of glioma recurrence. Therefore, combined assessment of oxygen metabolism, tissue hypoxia, microvascular architecture, and neovascularization activity might elucidate pathophysiological mechanisms involved in therapy resistance and glioma recurrence.

However, even advanced neuroimaging techniques for noninvasive in vivo detection of physiological imaging biomarkers are limited when approaching a combined investigation of these different processes. Consequently, multiple examinations with different neuroimaging modalities are required including techniques that are not well suited for in vivo investigations in humans due to their invasiveness (e.g., oxygen electrodes) or their limited availability and high costs (e.g., ^15^O positron emission tomography). This has hampered so far the implementation of such outcome measures in clinical studies on larger patient scale as well as for clinical routine use. In order to overcome these limitations, we previously presented a physiological magnetic resonance imaging (phyMRI) approach scoping to obtain and characterize quantitative characteristics about microvascular perfusion and architecture, neovascularization activity, and oxygen metabolism including tissue hypoxia in glioma patients [19,20,21,22]. The phyMRI technique combines vascular architecture mapping (VAM) [20] and quantitative blood oxygenation level–dependent (qBOLD) imaging [22] to obtain deeper insight into the brain tumors pathophysiology. The known physiological connection between neovascularization and tissue hypoxia [15] mainly drives the rationale for the combination of these two MRI methodologies.

We hypothesized that our phyMRI technique offers the capability to detect pathophysiological changes relevant and relative to the early developmental stage of AG recurrence in humans. The aims of this study were to determine pathophysiological alterations and the complex and dynamic interrelationship between physiological MRI biomarkers for microvascular architecture, neovascularization activity, and oxygen metabolism that precede radiological recurrence of IDH-mutated AGs.

## 2. Materials and Methods

### 2.1. Ethics

The study was approved and publicly registered by the Ethics Committee of the Lower Austrian Provincial Government (protocol code GS1-EK-4/339-2015, date of approval: 29 February 2016). The study was conducted in accordance with the guidelines of the Declaration of Helsinki. All included patients provided written informed consent prior to enrolment.

### 2.2. Patients

Patients were selected from an institutional glioma database that was prospectively populated between February 2016 and September 2020. Inclusion criteria were as follows: (i) age ≥18 years; (ii) WHO grading system based histopathological confirmation of an anaplastic glioma (AG, WHO grade III) with mutation of the IDH gene as initial diagnosis; (iii) treatment according to the standard of care, i.e., maximal safe and radical resection, radiotherapy, and concomitant and adjuvant chemotherapy with temozolomide [23]; (iv) follow-up MRI examinations with the study protocol; (v) no previous diagnosis of AG recurrence; (vi) no additional anti-glioma treatment but the standard of care (i.e., no antiangiogenic therapy etc.); and (vii) conventional MRI (cMRI) data were evaluated by at least two board-certified radiologists in consensus based on the updated Response Assessment in Neuro-Oncology (RANO) criteria [24,25].

### 2.3. MRI Data Acquisition

Follow-up MRI examinations were performed on a 3 Tesla whole-body scanner (Trio, Siemens, Erlangen, Germany; equipped with the standard 12-channel head coil) every 3–6 months or on an unscheduled basis in case of clinical signs of tumor recurrence. The cMRI protocol for clinical routine diagnosis of brain tumors included, among others, the following sequences: (i) an axial fluid-attenuated inversion-recovery (FLAIR) sequence; (ii) an axial diffusion-weighted imaging (DWI) sequence; (iii) pre- and post-contrast enhanced (CE) high-resolution three-dimensional (3D) T_1_-weighted magnetization-prepared rapid acquisition with gradient echo (MPRAGE) sequences; and (iv) a gradient echo (GE) dynamic susceptibility contrast (DSC) perfusion MRI sequence with 60 dynamic measurements during administration of 0.1 mmol/kg-bodyweight gadoterate-meglumine (Dotarem, Guerbet) at a rate of 4 mL/s using a MR-compatible injector (Spectris, Medrad). A 20-mL-bolus of saline was injected subsequently at the same rate. The parameters of the cMRI sequences are summarized in Table 1.

Microvascular architecture and neovascularization activity were investigated with the vascular architecture mapping (VAM) approach [20] by using a spin-echo DSC (SE-DSC) perfusion MRI sequence conducted with the same parameters and contrast agent injection protocol as described for the routine GE-DSC perfusion MRI (Table 1). Our approach to minimize patient motion and differences in time to first-pass peak, which may significantly affect the data evaluation, were described previously [21,22]. The SE-DSC perfusion MRI was performed before the GE-DSC perfusion MRI, because the SE-EPI technique is less sensitive to contrast-agent leakage [26]. This has the additional advantage that the first contrast agent injection for the SE-DSC perfusion MRI was like a pre-bolus for the more leakage-sensitive GE-DSC perfusion MRI.

Tissue oxygen metabolism and tension were investigated with the quantitative blood-oxygen-level-depended (qBOLD) imaging approach [27] using the following sequences: (i) a multi-echo GE sequence and (ii) a multi-echo SE sequence for mapping of the transverse relaxation rates R_2_* (= 1/T_2_*) and R_2_ (= 1/T_2_), respectively.

All phyMRI sequences for VAM and qBOLD carried out with identical geometric parameters (voxel size, number of slices, etc.) and slice position as used for the routine GE-DSC perfusion sequence (Table 1). The additional data acquisition time for the VAM (SE-DSC perfusion: 2 min) and qBOLD sequences (R_2_* and R_2_ mapping: 1.5 and 3.5 min) was seven minutes.

### 2.4. MRI Data Processing and Quantitative Analysis

Processing of cMRI, VAM, and qBOLD data as well as calculation of MRI biomarkers was performed with custom-made MatLab (MathWorks, Natick, MA) software. Processing of the cMRI data included calculation of the ADC = −ln[(S/S_0_)/b] from DWI data and calculation of absolute cerebral blood volume (CBV) and flow (CBF) maps from the GE-DSC perfusion MRI data via automatic identification of arterial input functions (AIFs) [28,29], respectively. This resulted in the MRI biomarker maps of microstructural density (ADC) and macrovascular perfusion (CBV and µCBV).

The VAM data processing pipeline (red lines in Figure 1) consisted of five steps: (i) correction for remaining contrast agent extravasation was performed as described previously [20,30,31]; (ii) fitting of the first bolus curves for each voxel of the GE- and SE-DSC perfusion MRI data with a previously described gamma-variate function [32], (iii) calculation of the ∆R_2,GE_ versus (∆R_2,SE_)^3/2^ diagram [33], the so-called vascular hysteresis loop (VHL) [20,34]. The VHL data were subsequently used for (iv) calculation of maps for phyMRI biomarkers of microvascular architecture including microvessel density (MVD), the vessel size index (VSI, i.e., microvessel radius) [35] as well as for neovascularization activity represented by the microvessel type indicator (MTI) [20]. For MVD and VSI, we used the following equations:(1)$$\mathrm{MVD}\;=\;\frac{Q_{\max}}{b}\cdot {\left(\frac{{\mathrm{CBV}}^{2}}{4\pi^{2}\cdot \mathrm{ADC}\cdot {\bar{R}}^{4}}\right)}^{1/3}$$
and
(2)$$\mathrm{VSI}\;=\;{\left(\frac{\mathrm{CBV}\cdot \mathrm{ADC}\cdot b^{3}}{2\pi \cdot Q_{\max}^{3}}\right)}^{1/2}$$
with Qmax = max[∆R_2,GE_]/max[(∆R_2,GE_)^3/2^]; $\bar{R}$ ≈ 3.0 μm is the mean vessel lumen radius and b is a numerical constant (b = 1.6781) [35]. MTI was defined as the area of the VHL signed with the rotational direction of the VHL, i.e., a clockwise VHL-direction was identified with a plus-sign, and a counter-clockwise VHL-direction was identified with a minus-sign [20]. In a final step (v) the map for the microvascular cerebral blood volume (μCBV) was calculated from the SE-DSC perfusion MRI data via a separate automatic identification of AIFs [28].

The qBOLD data processing pipeline (blue lines in Figure 1) consisted of three steps: (i) corrections for background fields of the R_2_*-mapping data [36] and for stimulated echoes of the R_2_-mapping data [37]; (ii) calculation of R_2_*- and R_2_-maps from the multi-echo relaxometry data. In the final step, (iii) MRI biomarker maps of oxygen metabolism including oxygen extraction fraction (OEF), cerebral metabolic rate of oxygen (CMRO_2_) [27], and the tissue oxygen tension (PO_2_) [38,39] were calculated using the following equations:(3)$$\mathrm{OEF}\;=\;\frac{R_{2}^{*}{-R}_{2}}{\frac{4}{3}{\cdot \pi \cdot \gamma \cdot \Delta \chi \cdot \mathrm{Hct}\cdot B}_{0}\cdot \mathrm{CBV}}$$
where γ (2.67502 × 10^8^ rad/s/T) is the nuclear gyromagnetic ratio; Δχ = 0.264 × 10^−6^ is the difference between the magnetic susceptibilities of fully oxygenated and fully deoxygenated haemoglobin; Hct = 0.42 × 0.85 is the microvascular hematocrit fraction, whereby the factor 0.85 stands for a correction factor of systemic Hct for small vessels;
(4)$${\mathrm{CMRO}}_{2}\;=\;\mathrm{OEF}\cdot \mathrm{CBF}\cdot C_{a}$$
where Ca = 8.68 mmol/mL is the arterial blood oxygen content [40]; and
(5)$${\mathrm{PO}}_{2}{\;=\;P}_{50}\sqrt[{h}]{\left(\frac{2}{\mathrm{OEF}}-1\right)}\;-\;\frac{{\mathrm{CMRO}}_{2}}{L}$$
where P_50_ is the hemoglobin half-saturation tension of oxygen (27 mmHg), h is the Hill coefficient of oxygen binding to hemoglobin (2.7), and L (4.4 mmol/Hg per minute) is the tissue oxygen conductivity as defined by Vafaee and Gjedde [41].

In summary, the data processing procedures resulted in two cMRI biomarker maps for microstructural density (ADC) and macrovascular perfusion (CBV) as well as six phyMRI biomarker maps for microvascular perfusion (µCBV), microvascular architecture (MVD and VSI), neovascularization activity (MTI), and oxygen metabolism (OEF, CMRO_2_, and PO_2_). All nine MRI biomarker maps are summarized at the bottom of Figure 1 (adapted from [42]). The time for processing the VAM (3.5 min) and qBOLD data (3.5 min) was seven minutes).

A quantitative analysis of the dynamic changes of the MRI biomarker data was conducted separately for both the subgroup of patients with recurrence of the AG and the subgroup without any radiological signs for recurrence, respectively. For the recurrence subgroup, this was executed in four steps starting firstly with definition of regions of interest (ROIs) on the CE T_1_-weighted MRI of the follow-up examination with clear radiological features of AG recurrence, i.e., the endpoint of the study period for a patient in this subgroup. Secondly, these ROIs were transferred to the previous follow-up MRI data that were performed during a one-year period before radiological AG recurrence. In a third step, MRI biomarker maps were coregistered to the respective CE T_1_-weighted MRIs, the ROIs were copied, and the mean MRI biomarker values were computed. Finally, the MRI biomarker values were plotted against the time difference between the date of the previous follow-up MRI and the date of follow-up MRI with clear features of recurrence, i.e., the time before radiological recurrence (TBRR; in days). This resulted in the time courses of the dynamic changes in the nine physiological parameters that occurred prior to recurrence of AG after standard therapy. For the patient subgroup without radiological recurrence during the study period, a one-year period prior to the most recent follow-up examinations was selected and evaluated as described above, with the difference that the ROIs were defined in signal alterations in the vicinity of the resection cavity. The parameter for the time axis was termed “time before recent follow-up” (TBRF; in days).

### 2.5. Statistical Analysis

Statistical analyses were performed using SPSS (version 21, IBM, Chicago, IL, USA) and R (version 3.6.3, R Foundation, Vienna, Austria). For the scatter plots of MRI biomarker values versus TBRR or TBRF, respectively, nonparametric regressions were performed using locally estimated scatterplot smoothing (LOESS) functions in order to obtain smooth curves. The LOESS trend lines were graphically analyzed for dynamic physiological changes that precede AG recurrence or occurred during stable disease, respectively. A two-sample Kolmogorov–Smirnov test was used to search for significant differences in the biomarker distribution between the two subgroups with and without AG recurrence during the study period. Furthermore, the MRI biomarker values were quarterly pooled for further statistical analysis. Significance of differences in quarterly pooled MRI biomarker values between the subgroups was calculated using a Mann–Whitney *U* test. Significance of differences in MRI biomarkers between the four quarters and the most recent follow-up (with or without radiological recurrence) within the two subgroups was determined using the one-way analysis of variance (ANOVA) method. The Tukey test was used as post-hoc procedure to be consistent with the assumption that homogeneity of variance was met and for correction for multiple comparisons. Homogeneity of variance was verified using the Levene’s test. When the assumption of homogeneity of variances was violated, Welch’s ANOVA in combination with the Games-Howell post-hoc test was used. *p* values less than 0.05 were considered to indicate significance.

## 3. Results

### 3.1. Patient Characteristics

The institutional database contained almost 1200 MR examinations using the study MRI protocol in 400 patients with brain tumors. A total of 60 patients (34 males; mean age 48.4 ± 11.9 years; 21–76 years), however, fulfilled the study inclusion criteria, 35 patients had the initial diagnosis of an anaplastic astrocytoma and 25 of an anaplastic oligodendroglioma WHO grade III with IDH gene mutation. From these 60 patients, 34 patients developed recurrence of the AG, and 26 patients showed no signs for AG recurrence during the study period. The mean time between resection of the AG and radiological recurrence was 34.8 ± 27.5 months (9.1–98.4 months). Treatment of the recurrent AG included a repeat craniotomy in 20 patients (59%, pathology confirmed recurrence in all cases; 7 patients revealed malignant transformation to glioblastoma with IDH gene mutation), repeat radiation therapy or repeat combined radio-chemotherapy in 6 patients (17%), a temozolomide rechallenge in 4 patients (12%), a second-line monotherapy with bevacizumab in 3 patients (9%) and with nivolumab in 1 patient (3%).

### 3.2. Follow-Up Examinations with Conventional and Physiological MRI

A total of 142 follow-up examinations of the 60 patients were evaluated in more detail. Calculation of the cMRI biomarker maps for microstructural density (ADC) and macrovascular perfusion (CBV) as well as the six phyMRI biomarker maps for microvascular perfusion (µCBV), microvascular architecture (MVD and VSI), neovascularization activity (MTI), and oxygen metabolism (OEF, CMRO_2_, and PO_2_) was successfully performed for all these follow-ups. The phyMRI biomarker maps provided indications for early changes in tissue physiology which occurred prior to tumor recurrence in patients with IDH-mutated AG.

Figure 2 shows an illustrative case demonstrating no signs for recurrence in cMRI (CE T_1_w, FLAIR, and CBV) during the first three follow-up MRI examinations (Figure 2A–C) but changes (red circles) in microvascular perfusion (µCBV) and microvascular architecture (MVD) and neovascularization activity (MTI) in the follow-up 105 days prior to radiological recurrence (Figure 2C). These alterations in microvasculature, however, were preceded by a local decrease in tissue oxygen tension (PO_2_), i.e., occurrence of local hypoxia (red arrows in Figure 2A,B). Neovascularization activity (MTI) and the associated increase in microvascular perfusion (µCBV) and architecture (MVD) leaded to a rebound in tissue oxygen tension (red arrows in Figure 2C,D). At radiological recurrence (Figure 2D), distinct contrast enhancement and increasing peritumoral edema were recognized, accompanied by increased perfusion (CBV and µCBV), microvascular architecture (MVD), and neovascularization activity (MTI) as well as a recovery of the tissue oxygen tension (increased PO_2_) to normal levels (and even higher).

Another representative case with tumor recurrence showed very strong local tissue hypoxia (low PO_2_) 208 days before radiological recurrence (Figure 3A). This hypoxic region might have been the starting point that initiated upregulation of neovascularization activity which was associated with increasing microvascular perfusion and architecture as well as beginning of recovery of tissue oxygenation approximately 4 months later (Figure 3B). Another three months later, cMRI revealed clear radiological features of recurrence, i.e., new contrast enhancement, increasing edema and hyperperfusion, as well as further progression of the microvascular alterations, increased neovascularization, and restored tissue oxygen tension (Figure 3C).

Figure 4 shows an illustrative case demonstrating no signs for recurrence (i.e., stable disease) during the whole study period for both cMRI and phyMRI.

### 3.3. Time Courses of MRI Biomarker Changes Preceding Radiological AG Recurrence

The MRI biomarker values obtained from 67 MRI examinations of the 34 patients who developed recurrence of the AG were plotted against the time before radiological recurrence (TBRR in days) of the tumor. The same procedure was carried out separately with the MRI biomarker data from 75 MRI examinations of the 26 patients without tumor recurrence during the study period. The resulting time courses provided information about the dynamic changes in perfusion, microstructural density, microvascular architecture, neovascularization activity, and oxygen metabolism that occurred before and during the development of AG recurrence (cyan data in Figure 5) as well as during a one-year period of stable disease (red data in Figure 5), respectively.

Both macrovascular (CBV) and microvascular perfusion (µCBV) revealed similar time courses prior to AG recurrence (Figure 5A,B). The values for CBV and µCBV showed a constant course until they increased from 180 and 200 days, respectively, before radiological recurrence, i.e., microvascular perfusion increased a little earlier compared to macrovascular perfusion. The density of microvessels (MVD), however, revealed a slight continuous decrease during the first six months and reached a minimum at about 180 days prior to radiological recurrence followed by a marked increase (Figure 5D). VSI, the second parameter for microvascular architecture (Figure 5E), however, showed a continuous increase over the whole one-year period prior to radiological recurrence. The time course of MTI (Figure 5F) was found to be rather similar to those for the perfusion parameters and demonstrated an upregulation of neovascularization activity 200 days prior to radiological AG recurrence, i.e., at about the same time as µCBV.

The time courses of the phyMRI biomarkers for oxygen metabolism, OEF and CMRO_2_, were found to be rather similar. In tissue which developed into recurrence of an AG, OEF (Figure 5G) showed an intensifying increase over the first six months and reached its maximum 180 days prior to radiological recurrence, i.e., increasingly more extraction of oxygen was required during this time period in order to meet the oxygen demand of the tissue. Interestingly, the OEF maximum was reached shortly after upregulation of neovascularization (Figure 5F) and at the same time as the increase in perfusion (Figure 5A,B). The maximum in OEF was followed by a strong decrease due to the development of new tumor vasculature enabling adequate blood supply and eventually hyperperfusion. The metabolic rate of oxygen (CMRO_2_, Figure 5H), however, reached the maximum about one month later than OEF and 150 days prior to radiological recurrence, i.e., CMRO_2_ showed a time lag in the course compared to OEF. Consequently, tissue oxygen tension (PO_2_; Figure 5I), which is influenced by both OEF and CMRO_2_, showed a continuous decrease (i.e., increasing tissue hypoxia) during the first six months that was even intensified at a TBRR of about 250 days and reached its minimum value (i.e., maximum tissue hypoxia) at about 150 days prior to radiological recurrence.

In summary, these time courses clearly demonstrated a close physiological interplay between the dynamics of blood perfusion, microvessel density, neovascularization activity, and oxygen metabolism in the one-year period prior to radiological recurrence of AG. The time period of 180 to 150 days before radiological recurrence seemed to be of special importance in recurrent AG development as it was associated with a switch from an avascular infiltrative to a vascularized proliferative phenotype of the tumor cells. The time course for microstructural density (ADC; Figure 5C) was supportive for this observation, because it reveal a decrease of microstructural density (increase in ADC) until 150 days prior to recurrence followed by an increase in microstructural density (decrease ADC) most probably due to tumor cell proliferation. The decrease in microstructural density during the first six months, however, might be associated with post-radiotherapeutic effect [43].

The time courses for the subgroup of patients without signs for AG recurrence during the study period (red data in Figure 5) revealed a constant course. For comparison purposes, the corresponding time courses for glioblastoma from a previous study [42] were also given in Figure 5 (dashed lines). These data provided interesting insights. The physiological changes associated with a recurrence of a glioblastoma are much more dynamic and stronger, especially for neovascularization activity and hypoxia. Therefore, the important time interval in which the phenotypic switch takes place during glioblastoma recurrence was around 150 to 120 days prior to radiological recurrence. In other words, the vascularized proliferative phase of recurrence till radiological detection was one month longer for AG when compared to glioblastoma.

### 3.4. Comparisons between Quarters

Statistical analysis of the quarterly pooled MRI biomarker data revealed that the most significant differences between the subgroups of patients with and without AG recurrence (marked by “*” in Figure 6), respectively, existed for CBV (one quarter before and at radiological recurrence), µCBV and MTI (two quarters and one quarter before radiological recurrence, and at radiological recurrence) as well as for CMRO_2_ and PO_2_ (two quarters before and at radiological recurrence). Significant differences between the quarters within the subgroup of patients with AG recurrence (marked by “†” in Figure 6) were found only between the last quarter before radiological recurrence and radiological recurrence itself for the following MRI biomarkers: CBV, µCBV, MVD, and MTI. There were no significant differences between the quarters within the subgroup of patients without AG recurrence.

## 4. Discussion

Tumor recurrence during the course of disease of patients with AG poses a diagnostic and therapeutic challenge. Although most radical repeat resection of the tumor is associated with significant survival advantages compared with no surgery or subtotal resection [44], early and reliable recurrence detection, which is an essential requirement for this treatment option, is often challenging even with advanced cMRI techniques such as CBV mapping [45]. Consequently, AG recurrence is detected at an advanced stage associated with progressive neurological deteriorations of the patient, potential malignant transformation towards glioblastoma, and limitations in treatment options.

The findings of this study support our hypothesis that early pathophysiological changes that occurred during the development of tumor recurrence in patients with IDH-mutated AG are detectable with our phyMRI approach. We searched for early alterations in macro- and microvascular perfusion, microvascular architecture, neovascularization activity, oxygen metabolism (including tissue hypoxia), and microstructural density that occurred during a one-year period prior to radiological recurrence of IDH-mutated AGs. Our main findings were threefold: (i) intensification of local tissue hypoxia starting at about 250 days prior to radiological recurrence initiated upregulation of neovascularization activity 50 to 70 days later and leaded to a switch from an avascular infiltrative to a vascularized proliferative phenotype of the tumor cells another 30 days later. (ii) The dynamic changes of blood perfusion, microvessel density, neovascularization activity, and oxygen metabolism showed a close physiological interplay in the one-year period prior to radiological recurrence of AG. (iii) phyMRI biomarkers for AG recurrence differed statistically significant from those for the stable disease as early as six months before radiological recurrence.

As mentioned above, studies investigating solely tumor recurrence in patient cohorts with IDH-mutated AGs are still lacking. The majority of the studies dealt with the prognostic value of molecular markers, especially the mutation of the IDH gene, indicating a favorable prognosis regarding progression free survival and time to recurrence associated with this finding [46,47,48,49,50]. The early alterations in phyMRI biomarkers associated with tumor recurrence—a decrease in microvessel density accompanied with an intensifying tissue hypoxia—might reflect in part the vascular cooption followed by formation of cuffs of tumor cells around microvessel as described previously in preclinical studies [15]. It is well-known that microvessels, which are coopted by glioma cells, express ANG-2, which in turn has been associated with endothelial cell apoptosis and vessel regression in absence of VEGF [17]. Furthermore, invading glioma cells displace the astrocytic endfeet from endothelial or vascular smooth muscle cells and disrupt the astrocyte-vasculature interaction that serve as exchange sites for ions, metabolites and energy substrates across the blood-brain-barrier [16]. Both mechanisms lead to a local decrease in tissue oxygen tension, i.e., local tissue hypoxia. These physiological alterations were macroscopically detectable with our phyMRI approach expressed as a decrease in MVD along with a continuously decreasing local tissue PO_2_.

Most recently, we investigated the pathophysiologic changes that precede tumor recurrence in 56 patients with IDH-wild type glioblastoma by using physiological MRI [42]. The time courses from this study were included as dashed lines for comparison purposes into Figure 5. The time courses of the phyMRI biomarkers were similar to those found in the current study. The initiating event in glioblastoma recurrence was also an intensification of local tissue hypoxia and a decrease in microvessel density. In glioblastomas, however, this was observed 190 days before radiological tumor recurrence compared to 250 days in the current study indicating significantly faster and temporal distinct dynamics of the physiological biomarkers changes due to increased aggressiveness of glioblastoma. Hence, the time frame between the first phyMRI-detectable alterations and radiological detection of tumor recurrence was two months longer in IDH-mutated AGs compared to IDH-wild type glioblastomas.

Loss in microvessel density and increase in tissue hypoxia (decrease in PO_2_) before initiation of neovascularization were more pronounced in glioblastoma compared to IDH-mutated AG, along with a stronger upregulation of neovascularization activity. Interestingly, the time interval between intensification of local tissue hypoxia and upregulation of neovascularization was about 70 days for both IDH-mutated AGs and IDH-wild type glioblastomas. This might be indicative that the biological processes during hypoxia-induced upregulation of neovascularization are independent of IDH gene mutation. Moreover, in both glioma entities a continued increase in tissue hypoxia (decrease in PO_2_) over another 30 days after the switch from an avascular infiltrative to a vascularized proliferative phenotype was observed. This continuation of local tissue hypoxia despite increasing neoangiogensis could be necessary for prolongation neovascular progression which otherwise might be downregulated by suppressive mechanisms. This time interval might be ideally suited to attack tumor recurrence at a very early stage via initialization of antiangiogenic [51], or antihypoxic therapy [52,53], or a combined treatment strategy targeting both angiogenesis and hypoxia [54].

Several limitations of our study need to be discussed. Firstly, our VAM approach requires two separate contrast agent injections. However, this ensures that the GE-DSC perfusion sequence, which is essential for clinical routine diagnosis, was kept unchanged regarding spatial and temporal resolution. Therefore, our approach allows the acquisition of VAM data with high signal-to-noise ratio, high spatial resolution, and coverage of the whole brain. The combined simultaneous GE-SE-DSC perfusion sequences [55,56,57] could be an alternative, but does not meet the above mentioned requirements for clinical routine diagnosis. Additionally, the GE-SE-approach usually uses a double dose or more [55,56,57]. Secondly, our multiparametric qBOLD approach provides only an estimation of the oxygen metabolism with model-inherent limitations. The model assumes that the system is in the static dephasing regime [58] whereby OEF is predominantly weighted to the medium sized and larger venules and provides an average blood oxygenation within the entire vasculature. Additionally, hemosiderin or protein accumulations, background gradients, white matter fiber orientation, and contrast agent leakage could bias the OEF estimation [59,60,61]. Finally, the number of patients was relatively small which was related to our rather strict inclusion/exclusion criteria (e.g., IDH mutation status, standard treatment only). Therefore, we performed no sub-analyses for differences in the time courses between anaplastic astrocytoma and anaplastic oligodendroglioma with IDH-mutation. Prospective studies evaluating the clinical usefulness of our phyMRI approach for AG recurrence detection deserve further attention.

## 5. Conclusions

Conclusively, we described the chronology of pathophysiological alterations that occur prior to tumor recurrence in patients with IDH-mutated AG after standard therapy using a non-invasive physiological MRI approach that is fully compatible with the requirements of radiological routine diagnostics of glioma. We detected significant changes in tissue hypoxia, microvascular architecture, neovascularization activity that occur up to six months before detection of recurrence with conventional MRI methods. These findings may help for implementing both new MR-based imaging modalities for routine follow-up monitoring of AG patients after standard therapy and may become useful to develop and implement novel tailored diagnosis and early therapy for recurrent AG. However, further preclinical as well as prospective in-human studies are required for validation, evaluation and application of our findings.

## Figures and Tables

**Figure 1 cancers-13-01797-f001:**
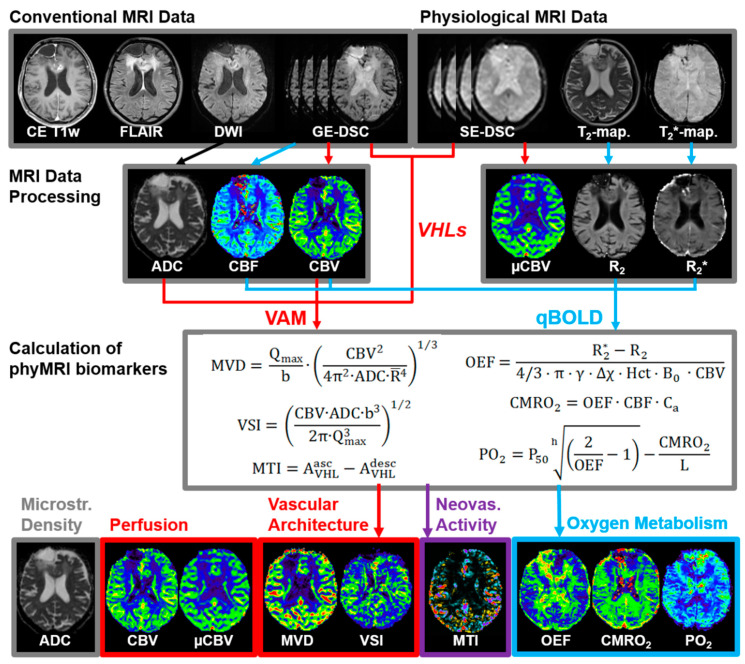
The pipeline for MRI data processing and calculation of MRI biomarker maps (adapted from [42]).

**Figure 2 cancers-13-01797-f002:**
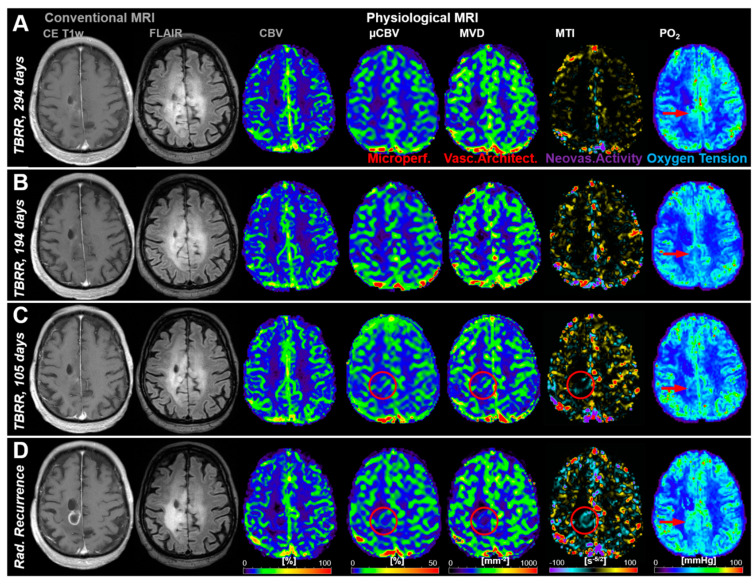
Follow-up MRI examinations of a 61-year-old male patient who developed recurrence of an anaplastic astrocytoma: (**A**) 294 days, (**B**) 194 days, and (**C**) 105 days, respectively, conventional anatomic MRI (CE T1w and FLAIR) and macrovascular perfusion (CBV) showed no indications for changes in the brain area where recurrence finally occurred. However, microvascular perfusion (µCBV), microvascular architecture (MVD), and neovascularization activity (MTI) revealed physiological changes (red circles) in the follow-up 105 days before radiological recurrence. Tissue oxygen tension (PO_2_), however, initially revealed a decrease from TBRR = 294 to 194 days (**A**,**B**), followed by a rebound till radiological recurrence. (**D**) Radiological recurrence was clearly detectable with conventional MRI and showed an increase in capillary perfusion and neovascularization activity. The patient received repeat combined radio-chemotherapy of the tumor. TBRR = time before radiological recurrence.

**Figure 3 cancers-13-01797-f003:**
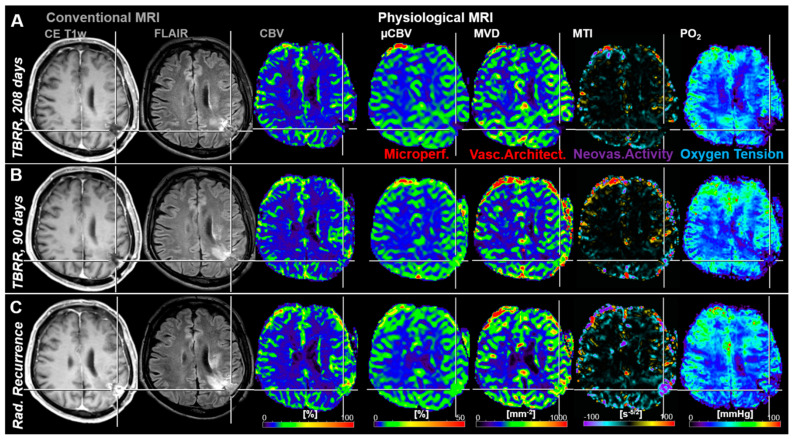
Follow-up MRI examinations of a 67-year-old male patient who developed local recurrence of an anaplastic astrocytoma (WHO Grade III). (**A**) The first follow-up examination (208 days before radiological recurrence) showed strong local hypoxia in the relevant region with low perfusion and microvascular architecture as well as without neovascularization activity. (**B**) In the second follow-up MRI examination, 118 days later and 90 days prior to radiological recurrence, a slight increase in microvascular perfusion, microvascular architecture, neovascularization activity, and local tissue oxygen tension was observed in this brain region. (**C**) This resulted in radiological recurrence detection of the AG another 90 days later. Repeat craniotomy revealed malignant transformation to glioblastoma. TBRR = time before radiological recurrence.

**Figure 4 cancers-13-01797-f004:**
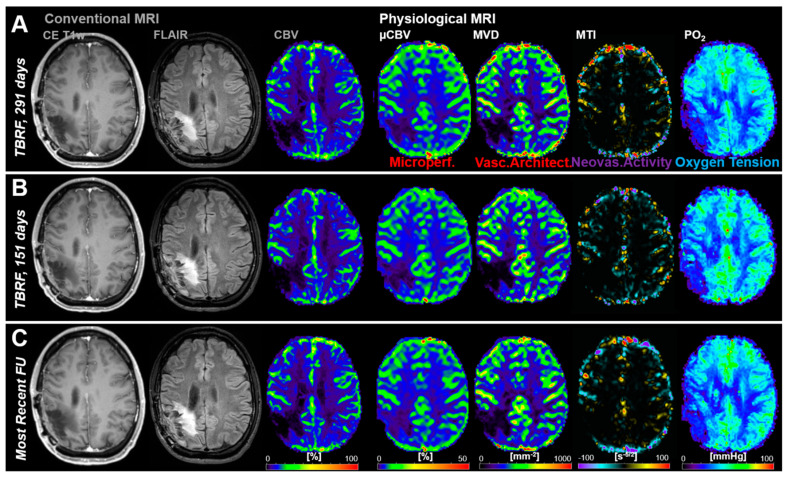
Follow-up MRI examinations of a 34-year-old male patient without recurrence of an anaplastic oligodendroglioma (WHO Grade III) during the study period. (**A**–**C**) Conventional MRI revealed no signs for radiological recurrence and physiological MRI biomarker maps showed no features for pathophysiological changes, respectively. TBRF = time before the recent follow-up (FU).

**Figure 5 cancers-13-01797-f005:**
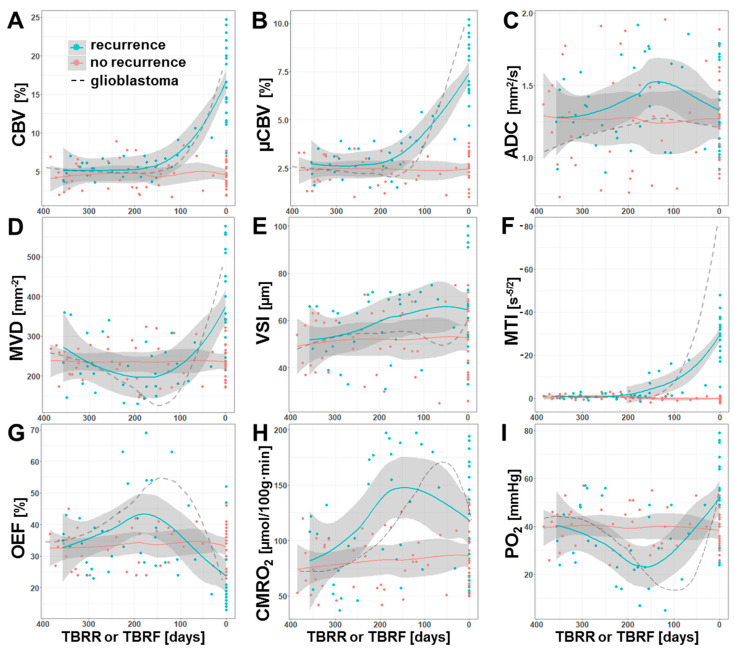
Time courses of the dynamic MRI biomarker changes that occurred prior to recurrence of an AG (cyan) or during a one-year-period without signs of recurrence (red), respectively. Scatter plots and locally-estimated-scatterplot-smoothing (LOESS) trend lines of MRI biomarker for (**A**) macrovascular perfusion (CBV), (**B**) microvascular perfusion (µCBV), (**C**) microstructural density (ADC), microvascular architecture represented by (**D**) microvessel density (MVD) and (**E**) diameter (VSI), (**F**) neovascularization activity (MTI), and oxygen metabolism (**G**: OEF; **H**: CMRO_2_, and **I**: PO_2_) versus time before radiological recurrence (TBRR, in days) in case of recurrence or versus time before most recent follow-up examination (TBRF, in days) in case of stable disease (no recurrence). The dashed lines, which are included for comparison purposes, represent the corresponding time courses for isocitrate-dehydrogenase (IDH)-wild type glioblastoma from a previous study [42].

**Figure 6 cancers-13-01797-f006:**
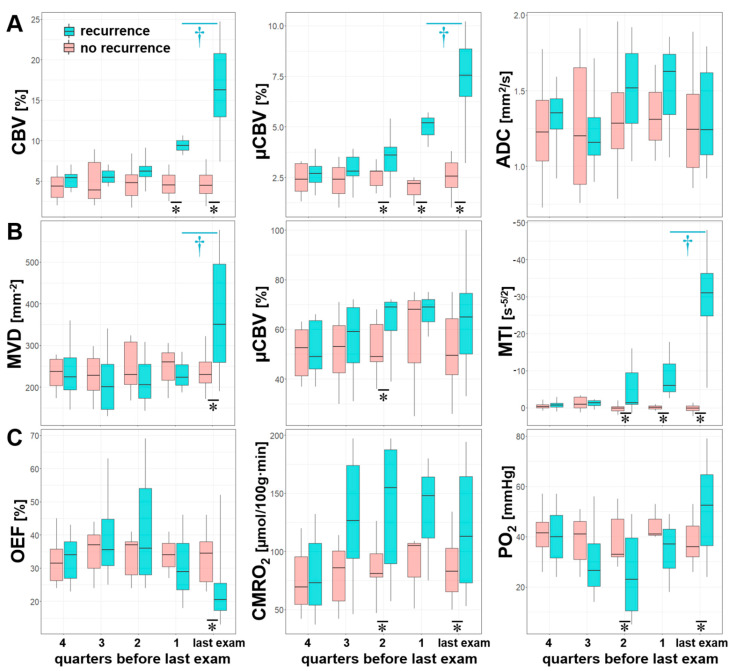
Series of box and whisker plots of quarterly pooled MRI biomarker of (**A**) perfusion (CBV and µCBV) and microstructural density (ADC), (**B**) microvascular architecture (MVD and VSI) and neovascularization activity (MTI), and (**C**) oxygen metabolism (OEF, CMRO_2_, and PO_2_) for AG recurrence (cyan) and stable disease (no recurrence, red), respectively. Crosses (†) mark significant differences between two successive quarters within the same patient subgroup, and asterisks (*) mark significant differences between patient subgroups within the same quarter (*p* < 0.05).

**Table 1 cancers-13-01797-t001:** Sequence parameters of the magnetic resonance imaging (MRI) study protocol.

	Conventional MRI Sequences				Physiological MRI Sequences		
	FLAIR	MPRAGE	DWI	GE-DSC	SE-DSC	R_2_ * Mapping	R_2_ Mapping
In-plane resolution	0.45 × 0.45	1.0 × 1.0	1.2 × 1.2	1.8 × 1.8	1.8 × 1.8	1.8 × 1.8	1.8 × 1.8
Slice thickness [mm]	3.0	1.0	4.0	4.0	4.0	4.0	4.0
Number of slices	48	176	29	29	29	29	29
TR [ms]	5000	2100	5300	1740	1740	1210	3260
TE [ms]	460	2.3	98	22	33	5–40 ms	13–104 ms
Flip angle * [°]	120	12	90	90	90	90	90
GRAPPA	2	2	2	2	2	2	2
other	TI = 1800 ms		b = 0 and 1000 s/mm^2^	60 dynamic volumes	60 dynamic volumes	8 echoes	8 echoes

FLAIR, fluid-attenuated inversion-recovery; MPRAGE, magnetization-prepared rapid acquisition with gradient echo sequence for contrast-enhanced T_1_-weoghted MRI; DWI, diffusion-weighted imaging; GE-DSC, gradient echo dynamic susceptibility contrast perfusion MRI; SE-DSC, spin echo dynamic susceptibility contrast perfusion MRI; GRAPPA, parallel imaging using generalized autocalibrating partially parallel acquisition. * Flip angle means the angle of excitation. Refocusing angles were 180° for all sequences with a SE scheme, i.e., FLAIR, DWI, SE-DSC, and R_2_ mapping.

## Data Availability

Data available on request due to privacy and ethical restrictions.

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
