# Peer review of "Hypoxia and Microvascular Alterations Are Early Predictors of IDH-Mutated Anaplastic Glioma Recurrence"

_cancers, 2021, doi:10.3390/cancers13081797_

Round 1
Reviewer 1 Report
This is a well-conducted prospective study on imaging biomarkers for early detection of recurrence in WHO grade 3, IDH mutant gliomas.
Minor comments:
- a shorter and catchier title would be desirable
- For the general reader, the term "physiological MRI" should be briefly introduced as a new and emerging technique
- Could you please indicate the time required for your phyMR programme ?
- page 7 , line 277: is it correct to mention "malignant transformation from WHO grade 3 IDH mut astrocytoma to glioblastoma" after treatment with radiation and chemotherapy ? Are these not molecular distinct tumor types and necrosis sequelae from former therapies ?
- page 10: since both macrovascular (CBV) and microvascular perfusion (μCBV) revealed similar time courses prior to glioma recurrence, will you proceed to assess both parameters ?
- legend figure 5 B: should read "micro" instead of "macro"
- In a former study you observed as an initiating event in recurrent IDH -wt glioblastoma an intensification of local tissue hypoxia and a decrease in microvessel density. In glioblastomas this was observed 190 days before radiological tumor recurrence whereas in the current study with WHO grade 3 IDH-mut glioma it was 250 days. Wouldn't you have expected a longer difference other than only two months ?
-
You mention that the observed time interval between the intensification of local tissue hypoxia and the upregulation of neovascularisation might be ideally suited to target tumour recurrence at a very early stage, for example with anti-angiogenic therapies. This is in contrast to what we do in clinical practice, where we give bevacizumab in late stage and obvious recurrence. How do you explain that bevacizumab had no benefit when given at an early stage, i.e. newly diagnosed glioblastoma ?
Reviewer 2 Report
This study retrospectively included 60 anaplastic gliomas (AG) patients after standard treatment (34 with recurrence and 26 without recurrence), and a total of 142 follow-up multiparametric MR images were evaluated for microvascular architecture and perfusion, neovascularization activity, oxygen metabolism, and hypoxia. The study design and presentation were similar to the authors' paper published in 2020 (reference 43) that focused on glioblastoma only. In this paper, they compared the differences of changes of these MR parameters between patients without and with recurrence, and also between anaplastic glioma and glioblastoma. Their results are quite interesting and provide new insights into the pathophysiology of tumor recurrence for AG. The data are well-presented and the figures are well-shown. The statistical analysis is appropriate. The conclusion is supported by the data.
Other specific comments:
1. Figure 1 is the same as Supplementary figure 1 of the previous paper (reference 43). Please refer to it.
2. Page 14: Please add the reference to the statement: "Most recently, we investigated the pathophysiologic changes that precede tumor recurrence in 56 patients with IDH-wildtype glioblastoma by using physiological MRI.".
3. Please describe the number of patients without and with tumor recurrence respectively in the abstract.
4. In this study, only patients with IDH mutation were included. In the introduction, the authors mentioned that "70 – 90% of AG show a mutation of the isocitrate dehydrogenase (IDH) gene". That means 10-30% of patients without IDH mutation were excluded in this study. Is there any specific reason to exclude these patients?
